# Zinc Ameliorates High Pi and Ca-Mediated Osteogenic Differentiation of Mesenchymal Stem Cells

**DOI:** 10.3390/nu16234012

**Published:** 2024-11-23

**Authors:** Enikő Balogh, Andrea Tóth, Dávid Máté Csiki, Viktória Jeney

**Affiliations:** MTA-DE Lendület Vascular Pathophysiology Research Group, Research Centre for Molecular Medicine, Faculty of Medicine, University of Debrecen, 4032 Debrecen, Hungary; andrea.toth@med.unideb.hu (A.T.); csiki.david.mate@gmail.com (D.M.C.); jeney.viktoria@med.unideb.hu (V.J.)

**Keywords:** zinc, human bone marrow-derived mesenchymal stem cells, osteogenic differentiation, RUNX2

## Abstract

Zinc is the second most abundant trace element in the human body, stored mainly in the bones. Zinc is required for bone growth and homeostasis and is also a crucial cofactor for numerous proteins that play key roles in maintaining microstructural integrity and bone remodeling. Bone marrow-derived mesenchymal stem cells (BMSCs) are multipotent progenitors found in the bone marrow stroma and can differentiate along multiple lineage pathways. In this study, we investigated the effect of zinc on the osteogenic differentiation of BMSCs. We stimulated the osteogenic differentiation of BMSCs with high phosphate and Ca-containing osteogenic medium (PiCa) in the presence or absence of zinc. We followed calcification by measuring ECM mineralization, the Ca content of the ECM, mRNA, and the protein expression of the osteo-chondrogenic transcription factor RUNX2 and SOX9 and its targets OCN and ALP. Zinc dose-dependently abolished PiCa-induced ECM mineralization and decreased the expression of RUNX2, SOX9, OCN, and ALP. Serum albumin did not alter the inhibitory effect of zinc on BMSC mineralization. Our further analysis with the zinc-chelator TPEN and ZnCl_2_ confirmed the specific inhibitory effect of free zinc ions on BMSC mineralization. Zinc inhibited phosphate uptake and PiCa-induced upregulation of the sodium-dependent phosphate cotransporters (PiT-1 and PiT-2). Zinc attenuated the PiCa-induced increase in ROS production. Taken together, these data suggest that zinc inhibits PiCa-induced BMSC calcification by regulating phosphate uptake and ROS production.

## 1. Introduction

Zinc is an important micronutrient that modulates numerous cellular processes including, but not limited to, DNA and protein synthesis, specific enzyme activities, and intracellular signaling [1]. It has been shown in recent years that bone homeostasis is also greatly influenced by zinc. Specifically, zinc increases bone mass both by inhibiting bone resorption by osteoclasts and by stimulating osteoblasts [2]. Because bone mass is typically maintained as a balance of osteoclastic bone resorption and osteoblastic bone formation [3], any disbalance between these processes results in metabolic bone diseases [4]. Highlighting the importance of zinc in bone formation, there is a growing body of evidence that shows that zinc deficiency leads to impaired skeletal growth in childhood [5,6,7] and an increased risk of osteopenia and osteoporosis in adulthood, regardless of gender [8,9]. Unsurprisingly, the zinc content of bone decreases with age, in the event of skeletal unloading, and also in postmenopausal conditions [10]. In spite of these observations in both animal models and human studies, the precise mechanisms by which zinc regulates bone formation at a cellular level remain poorly understood. In parallel with the loss of zinc from bone, tissue aging also leads to a decreased capability of mesenchymal stem cells (MSCs) to differentiate into osteogenic lineages [11].

BMSCs play a crucial role in bone homeostasis and are implicated in the pathophysiology of osteoporosis. These cells can differentiate into various cell types, including osteoblasts. In osteoporosis, the balance between bone formation and resorption is disrupted, leading to decreased bone density and increased fracture risk [12]. Several factors contribute to this imbalance, including alterations in BMSC function. As individuals age, the number and function of BMSCs decline, reducing their capacity to differentiate into osteoblasts and contribute to bone formation [13,14].

The effect of zinc on the function of key cells in bone metabolism has been examined in numerous studies. In the case of osteoblasts—which are crucial for bone formation—it has been shown that zinc deficiency (defined as <1 µmol/L) delays the induction of runt-related transcription factor 2 (RUNX2) expression by osteogenic stimuli, reduces extracellular matrix (ECM) mineralization, and decreases both cellular and matrix alkaline phosphatase (ALP) activity in murine, immortalized MC3T3-E1 preosteoblasts [15,16]. Regarding osteoclasts, it has been demonstrated that zinc inhibits the differentiation of macrophages into osteoclasts induced by the RANK ligand [17] and at high concentrations (>100 µmol/L) inhibits osteoclastic bone resorption under in vitro conditions [18].

In contrast, a recent study examined the effect of zinc on the osteoblastic differentiation of BMSCs, showing that zinc at high concentrations (100–200 µmol/L) enhances the osteoblastic differentiation of stem cells in a dose-dependent manner and also induces the expression of RUNX2 through a cAMP-PKA-CREB signaling pathway [19]. However, zinc has been shown to inhibit the osteogenic trans-differentiation of vascular smooth muscle cells (VSMCs) elicited by β-glycerophosphate, hydroxyapatite nanoparticles, and CaCl_2_ through the GPR39-dependent induction of TNFAIP3 and the consequent suppression of the NF-κB pathway [20]. Zinc showed similar anti-calcification results when applied in parallel with prolyl hydroxylase inhibitors, which can increase the calcification of VSMCs compared to high Pi and Ca applied alone [21]. As this process shares many aspects with BMSC osteogenic differentiation but is elicited using different stimuli in vitro, the current study explored the effects of zinc supplementation on osteogenic marker expression and matrix mineralization promoted by elevated Pi and Ca levels in BMSC cultures.

## 2. Methods

### 2.1. Materials

Unless otherwise specified, the reagents were purchased from Sigma-Aldrich (St. Louis, MO, USA).

### 2.2. Cell Culture

Human BMSCs derived from two different adult donors of Caucasian origin were obtained from ScienceCell Research Laboratories (Carlsbad, CA, USA). The cells were maintained in Dulbecco’s modified Eagle medium (DMEM, high glucose, D6171, Sigma) containing 10% FBS (10270-106, Gibco, Grand Island, NY, USA), antibiotic-antimycotic solution (A5955, Sigma), and L-glutamine (G7513, Sigma) and 1 mmol/L sodium pyruvate. Thew cells were maintained at 37 °C in a humidified atmosphere containing 5% CO_2_. The cells were grown until they reached confluence and used from passages 4 to 7. At confluence, BMSCs were switched to an osteogenic medium, which was prepared by adding inorganic phosphate 2.5 mmol/L Pi in the form of Na_2_PO_4_-Na_2_HPO_4_, pH 7.4, and 0.6 mmol/L Ca in the form of CaCl_2_ (C8106, Sigma) to the control medium in the presence or absence of zinc (1–100 µmol/L of ZnSO_4_ × 7H_2_O) or ZnCl_2_ (208086, Sigma) for 5 days. Alternatively, osteogenic differentiation was induced by an osteogenic medium, containing StemPro Osteogenesis Supplement (A1066-01, Gibco, Grand Island, NY, USA) and gentamicin (10 mg/mL). To evaluate the effects of zinc on BMSCs’ osteogenic differentiation, 50 µmol/L of ZnSO_4_ (Z2051, Sigma) and the specific zinc chelator N, N, N, N’-tetrakis (2-pyridinylmethyl)-1, 2-ethylenediamine (TPEN) (P4431, Sigma) were added to the culture media. The media were changed every 2 days.

### 2.3. Alizarin Red (AR) Staining and Quantification

After washing with PBS, the cells were fixed in 4% paraformaldehyde (16005, Sigma) and rinsed with deionized water thoroughly. The cells were stained with Alizarin Red S (A5533, Sigma) solution (2%, pH 4.2) for 10 min at room temperature. Excessive dye was removed by several washes in deionized water. To quantify AR staining in 96-well plates, we added 100 µL of hexadecylpyridinium chloride (C9002, Sigma) solution as the blank.

### 2.4. Quantification of Ca Deposition

The cells grown on 96-well plates were washed twice with PBS pH 7.4 without Ca^2+^ and Mg^2+^ and decalcified with HCl (30721, Sigma, 0.6 mol/L) for 30 min at room temperature. The calcium content of the supernatants was determined by a QuantiChrome Calcium Assay Kit (DICA-500, Gentaur, Kampenhout, Belgium). After decalcification, the cells were washed twice with PBS and solubilized with NaOH (0.1 mol/L) and sodium dodecyl sulfate (0.1%) solution. The protein content of the samples was measured with a BCA protein assay kit (Pierce Biotechnology, Rockford, IL, USA). The Ca content of the cells was normalized to the protein content and expressed as μg/mg protein.

### 2.5. Intracellular Phosphate Measurement

After 6 h of treatment, the cells were washed twice with 100 µL PBS and solubilized with 100 µL of 1% Triton-X 100. Cell lysates were assayed for inorganic phosphate using a QuantiChrome Phosphate Assay Kit (DIPI-500, Gentaur, Kampenhout, Belgium).

### 2.6. Quantification of OCN

For OCN detection, the BMSCs were grown and treated on six-well plates. Following the treatments, the cells were washed with PBS and then Ca was extracted from the ECM with 100 µL of ethylene-diamine-tetra-acetic acid (EDTA) (E6758, Sigma, 0.5 mol/L, pH 6.9). The OCN content of the EDTA-solubilized ECM samples was quantified by an enzyme-linked immunosorbent assay (ELISA) (DY1419-05, DuoSet ELISA, R&D, Minneapolis, MN, USA). All of the measurements were performed according to the manufacturer’s protocol. The OCN content was normalized to the protein content and expressed as ng OCN/mg protein.

### 2.7. Determination of Cell Viability

Cell viability was determined by the MTT assay. Briefly, the cells were washed with 100 µL of PBS, and 100 μL of 3-[4, 5-Dimethylthiazol-2-yl]-2, 5-diphenyl-tetrazolium bromide (0.5 mg/mL) solution in HBSS was added. After 4 h of incubation, the MTT solution was removed, formazan crystals were dissolved in 100 μL of DMSO, and optical density was measured at 570 nm.

### 2.8. Intracellular Zinc Detection

BMSCs were cultured with control or osteogenic media in the presence or absence of ZnSO_4_ (50 µmol/L) and TPEN (10 µmol/L) for up to 3 days. Intracellular zinc was detected using the zinc-selective FluoZin-3 AM probe (F24195, Invitrogen, Carlsbad, CA, USA), following the manufacturer’s instructions. Briefly, the cells were washed with PBS and loaded with FluoZin-3 AM (1 µmol/L, 30 min, at 37 °C). The cells were washed thoroughly, and fluorescence images were detected by excitation at 488 nm and emission at 542 nm under an Optika fluorescence microscope.

### 2.9. Quantitative RT-PCR

Total RNA was isolated from cells using TRIzol (RNA-STAT60, Tel-Test Inc., Friendswood, TX, USA) according to the manufacturer’s protocol. Two micrograms of RNA were reverse-transcribed to cDNA with High-Capacity cDNA Reverse Transcription Kits (Applied Biosystems, Waltman, MA, USA). Quantitative real-time PCR was performed using iTaq Universal Probes Supermix (Biorad Laboratories, Hercules, CA, USA). For the measurement of mRNA levels, the 5 µL reaction mixture contained 0.1 µg of the reverse-transcribed sample, 10 µmol/L of forward (5′-TACCCGCACTTGCACAAC-3′) and reverse (5′-CTCGCTCTCGTTCAGAAGTC-3′) for SOX9, forward (5′-GCATCCTATCAGTTCCCAATG-3′) and reverse primers (5-GAGGTGGTGGTGCATGGT-3′) for RUNX2, forward (5′-TGAATTCTGACGCCTCTGC-3′) and reverse (5′-GGTAGACACTCGGCAGCACT-3′) primers for PiT-1, forward (5′-TCCTCCTCAGACCGCTTTT-3′) and reverse primers (5-CCTGGTTCATCATCGCTAATC-3′) for HPRT, and 5 µL of SYBR Green Supermix (Bio-Rad, Hercules, CA, USA). PCRs were carried out using the Real-Time PCR System (Bio-Rad). Relative mRNA expressions were calculated with the ΔΔCt method using *HPRT* as the internal control.

### 2.10. Western Blot

To evaluate protein expression, BMSCs in six-well plates were lysed with 120 µL of Laemmli sample buffer. Then, 30 µL of the lysate was electrophoresed (100 V, 90 min) in SDS-PAGE (10%) and blotted onto a nitrocellulose membrane (10600002, Amersham Protran, GE Healthcare). Western blot analysis was performed with the use of an anti-Runx2 antibody (sc-390715) at a 1:200 dilution, anti-SOX9 antibody (PA5-81966, Invitrogen) at a 1:1000 dilution, anti-ALP antibody (sc-365765, Santa Cruz Biotechnology, Inc., Dallas, TX, USA) at a 1:500 dilution, anti-PiT-1 antibody (12423-1-AP, Proteintech, Chicago, IL, USA) at a dilution of 1:1000, and anti- PiT-2 antibody (12820-1-AP, Proteintech, Chicago, IL, USA) at a dilution of 1:1000. The antibodies were diluted in nonfat dry milk (1% in TBS-T) and incubated with the membranes overnight at 4 °C. The membranes were washed thoroughly with TBS-T. Then, the membranes were incubated (1 h) with horseradish peroxidase-labelled anti-rabbit or anti-mouse IgG secondary antibodies (NA-934 and NA-931 Amersham Biosciences Corp., Piscataway, NJ, USA) diluted at 1:800 in 1% nonfat dry milk in TBS-T and antigen-antibody complexes were detected by enhanced chemiluminescence using the Clarity^TM^ Western ECL Substrate (Bio-Rad Laboratories). Signals were detected by x-ray film or digitally using a C-Digit Blot Scanner (LI-COR Biosciences, Lincoln, NE, USA). After detection, the membranes were re-probed for β-actin using an anti-β-actin antibody at a dilution of 1:4000 (sc-47778, Santa Cruz Biotechnology, Inc., Dallas, TX, USA). The blots were quantified using the inbuilt software on the C-Digit Blot Scanner (LICOR Biosciences).

### 2.11. Intracellular ROS Measurement

The level of ROS was measured with a CM-H2DCFDA assay (Life Technologies, Carlsbad, CA, USA). The cells were loaded with the dye (10 μmol/L, 30 min) and then washed thoroughly with HBSS. After 3 days of treatment, the cells were washed with HBSS, and the fluorescence intensity was evaluated with the use of 488 nm excitation and 533 nm emission wavelengths. In some experiments, we applied the ROS inhibitor N-acetyl cysteine (NAC, 0.5 mmol/L) during the treatment. The experiments were repeated at least three times with five replicates.

### 2.12. Statistical Analysis

Data are presented as the mean ± SD with individual data points. Statistical analyses were performed with GraphPad Prism software (version 8.01). Comparisons between more than two groups were carried out by one-way analysis of variance (ANOVA) followed by Tukey’s multiple-comparison test. A value of *p* < 0.05 was considered significant.

## 3. Results

### 3.1. PiCa Induces the Mineralization of BMSCs in a Dose-Dependent Manner

The mechanisms of osteogenic differentiation in BMSCs and osteoblastic trans-differentiation in VSMCs exhibit similarities including common triggers. We exposed to BMSCs to an osteogenic medium containing different concentrations of excess phosphate (0–3.0 mmol/L) and Ca (0–1.2 mmol/L) for 5 days. Alizarin Red staining revealed that while Pi and Ca alone did not induce ECM mineralization, when these triggers were applied together, a dose-dependent mineralization of the ECM occurred, suggesting a synergistic effect of Pi and Ca in inducing calcification (Figure 1A,B). PiCa treatment did not influence cell viability, which was assessed by an MTT assay (Figure 1C). Based on these results, we used an osteogenic medium supplemented with 2.5 mmol/L Pi and 0.6 mmol/L Ca to induce the osteogenic differentiation of BMSCs throughout this paper.

### 3.2. Zinc Supplementation Inhibits the PiCa-Induced Mineralization of BMSCs in a Dose-Dependent Manner

Previously, zinc has been shown to prevent the phosphate (Pi)-induced osteogenic trans-differentiation of VSMCs [22]. To examine whether zinc inhibits Pi and the Ca-provoked mineralization of BMSCs in our experimental conditions, we cultured human BMSCs in an osteogenic medium containing 2.5 mmol/L Pi and 0.6 mmol/L Ca (PiCa), in the presence or absence of zinc (1–100 µmol/L), for 5 days. As revealed by Alizarin Red staining, zinc caused a dose-dependent attenuation of PiCa-induced mineralization, leading to complete inhibition from 50 µmol/L (Figure 2A). Next, we examined the effect of zinc on the Ca content of the ECM. Zinc caused a dose-dependent decrease in the Ca content of the ECM (Figure 2D). At the concentration of 50 µmol/L, zinc lowered the Ca content of the ECM of PiCa-stimulated cells down to the level of non-stimulated cells (Figure 2D). Subsequently, we checked whether the anti-osteogenic effect of zinc is selective for the PiCa-induced osteogenic differentiation of BMSCs or if it is a more general inhibitor of osteogenesis. We triggered osteogenic differentiation with an osteogenic basal medium (OBM) in the presence or absence of zinc. Alizarin Red staining revealed that zinc strongly inhibited the ECM mineralization of BMSCs triggered by OBM (Figure 2B). This result was confirmed by measuring the Ca content of the ECM (Figure 2E). To analyze whether the inhibitory effect of ZnSO_4_ is not due to its SO_4_^2−^ content, we used a different Zn source, ZnCl_2_. Like ZnSO_4_, ZnCl_2_ treatment also significantly inhibited calcium deposition in BMSCs at day 5 (Figure 2C,F). To determine whether any of the treatments are cytotoxic to BMSCs, we determined cell viability with an MTT assay. Our data revealed that none of the treatment combinations influenced cell viability (Figure 2G–I).

### 3.3. Zinc Down-Regulates the Expression of RUNX2, SOX9, and Its Downstream Targets ALP and OCN

VSMC differentiation to an osteogenic phenotype characterized by elevated levels of osteogenic markers has been identified as the underlying mechanism of vascular calcification [23]. Therefore, we assessed the effect of zinc on the expression of the key osteo-chondrogenic transcription factors, RUNX2 and SRY-box containing gene 9 (SOX9), and its downstream targets, osteocalcin (OCN) and ALP. PiCa stimulation induced a 1.4-fold increase in the RUNX2 mRNA level that was diminished by zinc (Figure 3A). Importantly, zinc at 50 µmol/L decreased the level of RUNX2 mRNA down to the level of unstimulated cells (Figure 3A). PiCa caused no measurable induction of RUNX2 at the protein level. In turn, zinc at the concentrations at and above 50 µmol/L completely abolished the protein expression of RUNX2 (Figure 3B). Parallel with these findings, PiCa stimuli can trigger a 15-fold increase in the SOX9 mRNA level, which was abrogated by zinc (Figure 3C). Western blot analyses confirmed the gene expression results (Figure 3D). Osteogenic stimulation triggers the trans-differentiation of BMSCs to osteoblast-like cells. This process can be monitored by the detection of certain osteoblast-specific proteins including OCN and ALP. Therefore, next, we investigated the effect of zinc on the expression of these markers. Next, we determined the level of OCN, a major non-collagenous protein found in the bone matrix, in the ECM of the BMSCs. The OCN level of PiCa-treated BMSCs was about 11-fold higher than that in the control. This increase was completely absent in the presence of zinc at a concentration of 50 and 100 µmol/L (Figure 3E). PiCa triggered a roughly 1.6-fold increase in the protein expression of ALP as compared to the control BMSCs (Figure 3F). This increase was diminished in the presence of zinc at the concentrations of 5, 25, 50, and 100 µmol/L.

### 3.4. Zinc Chelator-TPEN Abrogates the Inhibitory Effect of Zinc in BMSCs

It has been reported that serum albumin is the major zinc carrier in the blood and is responsible for its systemic distribution Approximately 75–80% of plasma zinc is bound to albumin [24]. To evaluate whether albumin influences the zinc-mediated inhibition of the osteogenic differentiation of BMSCs, we cultured the cells in PiCa in the presence or absence of zinc (50 µmol/L) and albumin (0.01–1 mg/mL). Alizarin Red staining and quantitative calcium analysis showed that zinc significantly attenuated calcium deposition in BMSCs at day 5 and albumin did not alter the inhibitory effect of zinc on BMSCs mineralization in any concentration range (Figure 4A,B). Previous studies showed that zinc inhibits human valve interstitial cell calcification in vitro, and this inhibitory effect is eliminated by the zinc chelator TPEN [25]. Therefore, to investigate whether the inhibitory effect of zinc on BMSCs osteogenesis was specifically due to free zinc ions, a specific intracellular zinc chelator TPEN was used. Alizarin Red staining, quantitative calcium analysis, and the OCN level revealed that 10 µmol/L of TPEN significantly reversed the inhibitory effect of 50 µmol/L of zinc (Figure 4C–E). Taken together, these data suggest that non-chelated zinc inhibits BMSC mineralization.

Following this, we investigated the effect of TPEN on the zinc-induced downregulation of osteogenic markers in PiCa-stimulated BMSCs. Zinc at 50 µmol/L decreased RUNX2, SOX9, and ALP protein expression well below that of control cells. TPEN mitigated the inhibitory effect of zinc on the osteogenic stimuli-induced upregulation of RUNX2, SOX9, and ALP (Figure 5A). FluoZin-3 (a zinc-selective indicator) staining revealed that 10 µmol/L TPEN decreased the zinc accumulation in BMSCs treated with 50 µmol/L of zinc at day 3 (Figure 5B).

### 3.5. Zinc Prevents PiCa-Mediated Phosphate Uptake

Evidence suggests that the effects of hyperphosphatemia are mediated via PiT-1, which facilitates the entry of Pi into vascular cells [26]. To further explore the mechanism by which zinc inhibits the mineralization of BMSCs, we measured the Pi uptake of BMSCs in the presence or absence of zinc. Intriguingly, the addition of zinc inhibited Pi uptake in a dose-responsive manner, providing significant and complete suppression at concentrations of 50 μmol/L and 100 μmol/L, respectively (Figure 6A). To explore the mechanism underlying the inhibition of Pi uptake, we examined the expression of PiT-1. PiCa treatment induced a 1.6-fold elevation in the PiT-1 mRNA level in BMSCs (Figure 6B). Zinc prevented a PiCa-induced increase in sodium-dependent phosphate cotransporter (PiT-1 and PiT-2) expressions (Figure 6C). These results suggest that zinc induces a decrease in the phosphate uptake of BMSCs which may contribute to the anti-osteogenic effect of zinc.

### 3.6. Zinc Prevents PiCa-Mediated ROS Production

It has been shown that zinc supplementation decreases reactive oxygen species (ROS) production in vitro [27] through the upregulation of ROS scavenger proteins, which mechanism was proposed in the zinc-induced inhibition of VSMC calcification [28,29,30]. Considering that zinc affects ROS generation and that ROS are recognized as a critical factor in the osteogenic differentiation of BMSCs, we next examined the effect of osteogenic stimuli on ROS levels in the presence or absence of an ROS scavenger NAC and zinc. PiCa stimulation increased ROS generation in BMSCs compared to non-stimulated cells (Figure 7A). NAC at the dose of 0.5 mmol/L completely inhibited the PiCa-mediated excessive production of ROS (Figure 7A,B). Then, we investigated whether scavenging excess ROS can prevent PiCa-induced BMSC calcification. Alizarin Red staining revealed that mineralization did not occur in the presence of NAC (0.5 mmol/L) (Figure 7B). Finally, we observed that zinc, similar to NAC, inhibits the PiCa-induced increase in ROS production (Figure 7C), which could explain the anti-osteogenic effect of zinc.

## 4. Discussion

In this study, we used BMSCs to investigate the effect of zinc on the osteogenic differentiation process. During BMSC differentiation towards a specific cell type, numerous stimuli and inhibitors are crucial for both the initial commitment and the later stages of differentiation. The differentiation of BMSCs into distinct cell types is regulated by various cytokines, growth factors, extracellular matrix molecules, and transcription factors [31]. Osteogenic medium supplemented with dexamethasone, bone-morphogenetic proteins, β-glycerolphosphate, L-ascorbic acid-2-phosphate, and combinations of transforming growth factor-beta (TGF-β) and vitamin D has been commonly used to induce the osteogenic differentiation of BMSCs in vitro [32,33]. In our experiments, we opted to induce the osteogenic differentiation of BMSCs using high Pi and Ca, which are well-established and pathophysiologically relevant triggers of the osteoblastic trans-differentiation of VSMCs [34]. We observed that elevated Pi and Ca promoted ECM mineralization, increased Ca content, and Alizarin Red positivity. ECM mineralization occurs significantly faster in response to Pi and Ca (Figure 1) when compared to the previously described inducers (5 days vs. 8–14 days), which could be a great advantage in bone tissue engineering.

The effects of zinc on osteoblasts and osteoclasts have been explored in previous studies, revealing that zinc plays a crucial role in cell growth and proliferation. It has been demonstrated that zinc enhances bone formation and mineralization, reduces bone resorption, and stimulates ALP activity. Additionally, zinc has been found to inhibit osteoclastic bone resorption in vitro [35,36].

Bone homeostasis is regulated by several coordinated cellular and molecular processes, including bone formation and resorption, as well as the synchronized activity of osteoblasts and osteoclasts [37]. Osteoblasts, responsible for bone formation, differentiate from MSCs during bone remodeling. A recent study showed that zinc enhances, rather than decreases, the osteogenic differentiation of MSCs in a dose-dependent manner [19], albeit under different experimental conditions than those used in the current experiments. Specifically, zinc was shown to have this effect at higher concentrations (100–200 μmol/L) and using a different osteogenic stimulus (low glucose, 10 mM β-glycerophosphate, 100 nM dexamethasone, and 50 μg/mL ascorbic acid). However, another research group found that zinc inhibits the osteoblastic trans-differentiation of VSMCs [20]. These two studies somewhat contradict each other, as the osteogenic differentiation of BMSCs and the osteoblastic trans-differentiation of VSMCs share many similarities. Cell type-dependent differences could explain the opposite osteogenic responses observed among these cells following zinc treatment. In contrast to zinc, iron has been shown to affect the osteogenic differentiation of both VSMCs and BMSCs in the same manner, causing the inhibition of mineralization [38,39].

In this study, we confirmed that zinc is a potent inhibitor of the osteogenic differentiation of BMSCs. Zinc attenuated the osteogenic stimuli (PiCa or OBM)-induced ECM mineralization of BMSCs (Figure 2). In response to PiCa stimuli, BMSCs increase the expression of SOX9 and RUNX2, the key transcription factors in the osteo-chondrogenic differentiation process. RUNX2 has been established in multiple previous studies as a master regulator of osteoblast differentiation, both in vitro and in vivo [40]. RUNX2 activation is among the most important temporal events occurring during the osteogenic differentiation of BMSCs. This is supported by the fact that RUNX2-deficient mice completely lack mature osteoblasts and mineralized bone formation [41]. Although RUNX2 initiates the expression of bone matrix protein genes in MSCs as they enter osteoblastic differentiation, it blocks the final maturation steps of osteoblasts [42]. In a murine osteoblast cell line, zinc supplementation increases RUNX2 expression at the mRNA and protein levels, while zinc deficiency reduces RUNX2 expression and the nuclear RUNX2 protein levels during osteogenic differentiation induced by 10 mmol/L glycerol 2-phosphate and 50 μg/mL ascorbic acid [16]. Given the central role of RUNX2 in osteogenic differentiation and the evidence that zinc likewise may influence osteogenesis, we hypothesized that zinc might contribute to the regulation of RUNX2 during the osteogenic differentiation of BMSCs.

Zinc completely abrogated the PiCa-induced upregulation of RUNX2 and SOX9. Parallel with this, PiCa-induced elevations of OCN and ALP expressions were attenuated in the presence of zinc (Figure 3). Zinc, an essential trace element for maintaining normal health, has been linked to various human diseases. To the best of our knowledge, this study is the first to report that zinc supplementation significantly inhibits the mineralization of BMSCs. In healthy individuals, zinc bioavailability is affected by several factors, including the individual’s zinc status, the overall zinc concentration, and the availability of soluble zinc in the diet [43]. We demonstrated that 50 µM of zinc inhibited the mineralization of BMSCs. Although the optimal serum zinc levels in adults are typically maintained between 13.8 and 22.9 µM [44], most of the body’s zinc is stored in bone tissue; therefore, local concentrations are most likely higher than what can be measured in the blood. Furthermore, zinc is predominantly carried by serum albumin in the blood [24]. The addition of albumin did not alter the inhibitory effect of zinc on BMSCs mineralization (Figure 3). Further experiments using TPEN chelator ( Figure 4; Figure 5) and ZnCl_2_ (Figure 2) confirmed that zinc specifically inhibits the BMSCs mineralization process. Phosphate uptake via the sodium-dependent phosphate cotransporters, PiT-1 and PiT-2, is crucial for VSMC calcification and phenotypic modulation in response to increased phosphate levels [45]. Intriguingly, the addition of zinc inhibited Pi uptake in a dose-responsive manner, providing substantial and complete suppression at concentrations of 50 μmol/L and 100 μmol/L, respectively (Figure 6). To explore the mechanism underlying the inhibition of Pi uptake, we examined the expression of PiT-1 and PiT-2. PiCa treatment induced a mild elevation in the expression of both phosphate transporters. Zinc treatment decreased the expression of PiT-1 and PiT-2 below the level of unstimulated cells (Figure 6C).

There is evidence for the role of ROS in the survival, proliferation, and differentiation of BMSCs. ROS influence osteogenesis by modulating various signaling pathways, including FOXO, Wnt, and Hedgehog signaling in BMSCs [46]. Several studies have indicated a connection between oxidative stress, osteogenic differentiation, and bone formation. Oxidative stress is known to diminish the osteogenic differentiation of murine preosteoblastic cell lines (MC3T3-E1) as well as bone marrow-derived stromal cells (M2-10B4) [47]. Numerous studies have demonstrated a negative correlation between zinc availability and oxidative stress. For example, zinc deficiency has been shown to increase the production of ROS and inflammatory cytokines (TNF-α, IL-1β, and IL-8). Zinc is a potent antioxidant and anti-inflammatory agent, and proper zinc supplementation can reduce oxidative stress and inflammatory responses [48]. In this study, we investigated the role of ROS in the PiCa-induced osteogenic differentiation of BMSCs. We found that the PiCa-induced osteogenic differentiation of BMSCs is accompanied by an increased generation of ROS. The PiCa-induced ROS production and osteogenic differentiation of BMSCs were inhibited by the glutathione-precursor NAC, suggesting a causative role of ROS in the PiCa-induced osteogenic differentiation of BMSCs (Figure 7A,B). Additionally, we found that zinc inhibits PiCa-induced ROS generation, which may be responsible for the anti-osteogenic effect of zinc (Figure 7C).

## 5. Conclusions

In conclusion, our findings provide evidence that excess zinc specifically inhibits the in vitro osteogenic differentiation of BMSCs via the downregulation of the key osteo-chondrogenic transcription factors RUNX2 and SOX9 and its target genes OCN and ALP and inhibits ECM mineralization. It is important to note that the validation of our results with in vivo experiments would greatly increase their translational potential. We provided two mechanisms—(i) the zinc-induced reduction in phosphate uptake and (ii) the zinc-induced attenuation of ROS production—as potential contributors to the anti-osteogenic effect of excess zinc.

## Figures and Tables

**Figure 1 nutrients-16-04012-f001:**
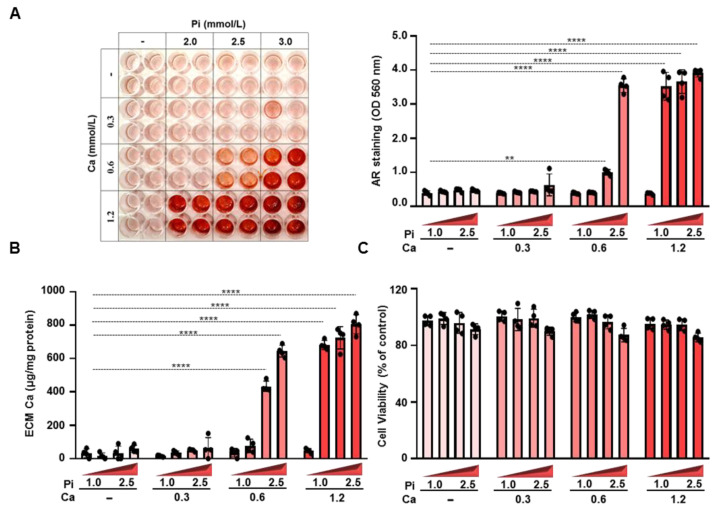
**PiCa induces the mineralization of BMSCs in a dose-dependent manner.** Confluent BMSCs (passages 4–6) were cultured under control (Ctrl) or osteogenic conditions containing excess phosphate (Pi, 0–3.0 mmol/L) and excess Ca (0.3–1.2 mmol/L). (**A**) Ca deposition as a readout of ECM (day5) mineralization was visualized by Alizarin Red staining. Representative images and quantification are depicted from three independent experiments. (**B**) Ca content of HCl-solubilized ECM. (**C**) Cell viability was measured by an MTT assay. Data are expressed as the mean ± SD, n = 4. Ordinary one-way ANOVA followed by Tukey’s multiple comparison test was used to calculate *p* values. ** *p* < 0.01, **** *p* < 0.001.

**Figure 2 nutrients-16-04012-f002:**
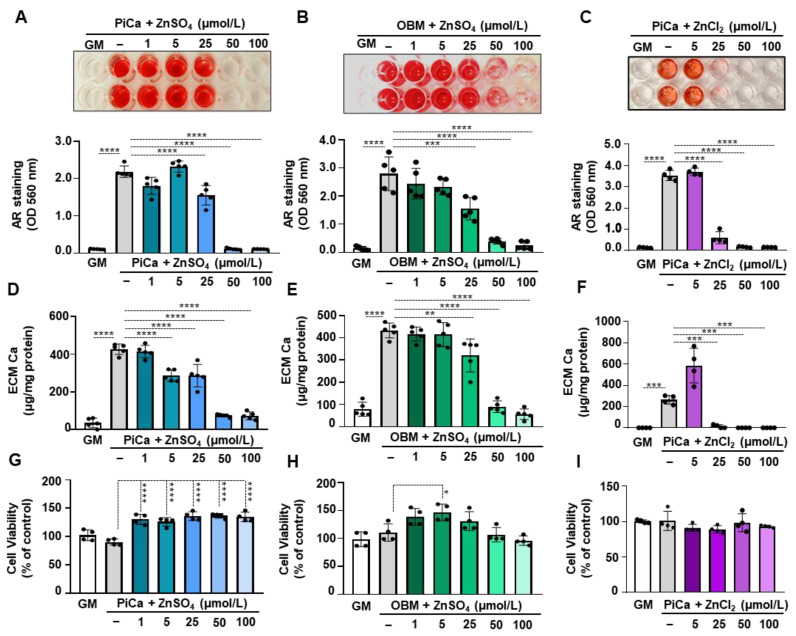
**Zinc inhibits the PiCa-induced mineralization of BMSCs in a dose-dependent manner.** Confluent BMSCs (passage 4–6) were cultured under control (Ctrl) or osteogenic conditions (2.5 mmol/L Pi, and 0.6 mmol/L Ca, PiCa) or OBM medium in the presence or absence of ZnSO_4_ or ZnCl_2_ (1–100 µmol/L) for 5 days. (**A**–**C**) Ca deposition as a readout of ECM mineralization was visualized by Alizarin Red staining. Representative images of stained plates are shown (n = 4–5). (**D**–**F**) The Ca content of HCl-solubilized ECM is shown. Results are expressed as the mean ± SD, n = 4–5. (**G**–**I**) Cell viability was determined by an MTT assay after 5 days of treatment. Data represent the mean ± SD, n = 4–5. Ordinary one-way ANOVA followed by Tukey’s multiple comparison test were used to calculate *p* values. * *p* < 0.05, ** *p* < 0.01, *** *p* < 0.005, **** *p* < 0.001. Two additional experiments yielded similar results.

**Figure 3 nutrients-16-04012-f003:**
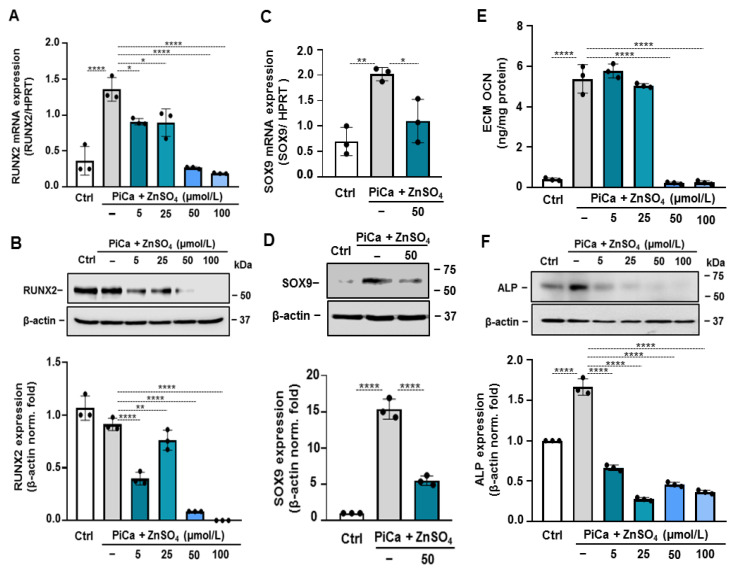
**Zinc inhibits the PiCa-induced upregulation of RUNX2, SOX9, OCN, and ALP.** Confluent BMSCs (passage 6–8) grown in six-well plates were treated with osteogenic stimuli supplemented with zinc at the indicated concentrations (0–100 µmol/L) for 3 days. (**A**,**C**) RUNX2 and SOX9 mRNA levels were determined by real-time RT-PCR. (**B**,**D**,**F**) Protein expression of RUNX2, SOX9, and ALP was determined from whole cell lysates. Membranes were re-probed for β-actin. Representative Western blots and the relative expression of RUNX2, SOX9, and ALP normalized to β-actin from three independent experiments are shown. Results are presented as the mean ± SD of three independent experiments performed in triplicate. * *p* < 0.05, ** *p* < 0.01, **** *p* < 0.001. (**E**) OCN levels were determined in EDTA-solubilized ECM samples by ELISA (day 5). Results are presented as the mean ± SD of three independent experiments performed in triplicate.

**Figure 4 nutrients-16-04012-f004:**
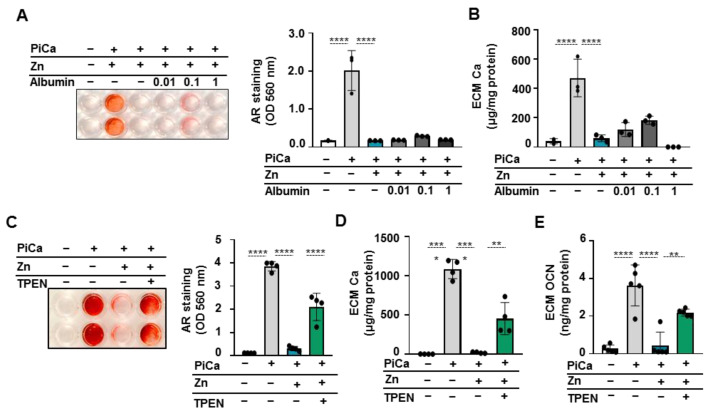
**Albumin does not affect the inhibitory effect of zinc, but the zinc chelator TPEN decreases the inhibitory effect of Zinc in BMSCs.** BMSCs (passage numbers 5–8) were cultured in Ctrl or osteogenic medium in the presence or absence of zinc (50 μmol/L) and TPEN (10 µmol/L). (**A**,**C**) Representative Alizarin Red staining (day 5) and quantification. (**B**–**D**) The Ca content of the HCl-solubilized ECM is presented. (**E**) OCN level in EDTA-solubilized ECM samples (day 5). Data are expressed as the mean ± SD. * *p *< 0.05, ** *p *< 0.01, *** *p *<0.005, **** *p *< 0.001.

**Figure 5 nutrients-16-04012-f005:**
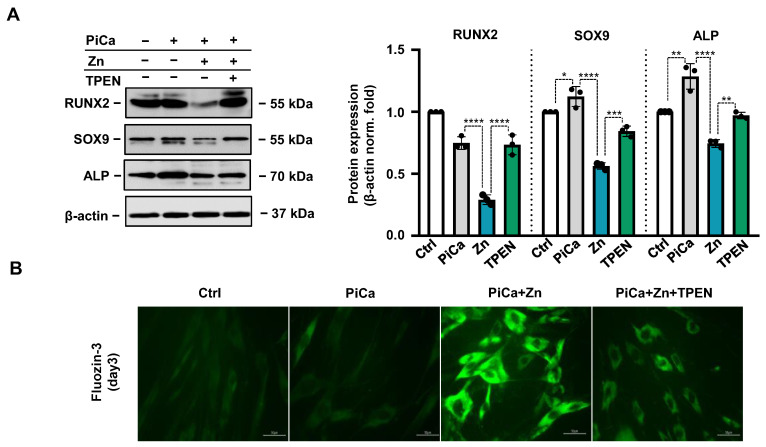
**TPEN attenuates the inhibitory effect of zinc on the PiCa-induced upregulation of RUNX2, SOX9, and ALP and prevents intracellular zinc accumulation in BMSCs.** (**A**) RUNX2, SOX9, and ALP expressions were determined from whole cell lysates (day 3). Membranes were re-probed for β-actin. Representative Western blots and relative expression of RUNX2, SOX9, and ALP normalized to β-actin from three independent experiments are shown. * *p* < 0.05, ** *p* < 0.01, *** *p* < 0.005, **** *p* < 0.001. (**B**) Representative images of Fluozin-3 staining at day 3. Scale bar 50 µm. Data are expressed as the mean ± SD of three independent experiments performed in triplicate.

**Figure 6 nutrients-16-04012-f006:**
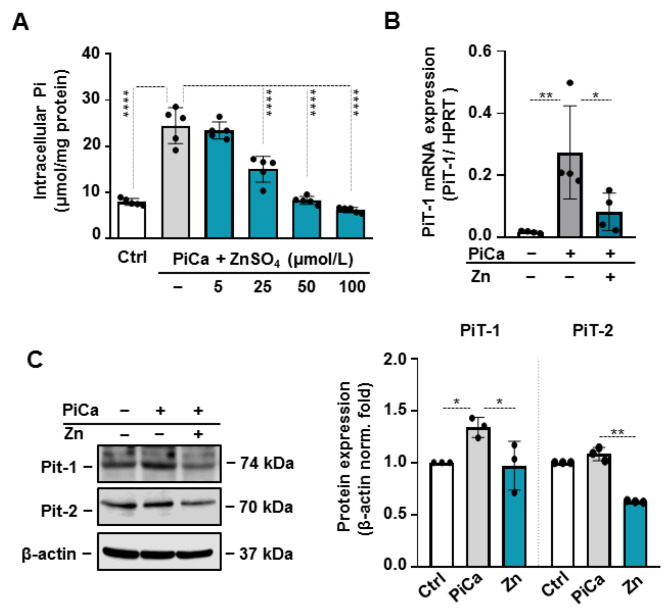
**Zinc inhibits phosphate uptake by BMSCs.** BMSCs were cultured in Ctrl or PiCa medium alone or supplemented with zinc (5, 25, 50, and 100 µmol/L) for 6 h. (**A**) The intracellular phosphate content of cell lysates was determined with a phosphate assay kit (n = 5). Data are expressed as the mean ± SD. **** *p *< 0.001. (**B**) BMSCs were cultured in Ctrl and PiCa medium alone or supplemented with zinc (50 µmol/L) for 48 h and the PiT-1 mRNA level was determined. Data are expressed as the mean ± SD, n = 4. (**C**) The protein expression of type III sodium-dependent phosphate cotransporter 1 (PiT-1 and PiT-2) in whole cell lysates (48 h). Membranes were re-probed for β-actin. Representative Western blots and quantification of PiT-1 and PiT-2 relative expressions are shown (n = 3). * *p* < 0.05, ** *p *< 0.01.

**Figure 7 nutrients-16-04012-f007:**
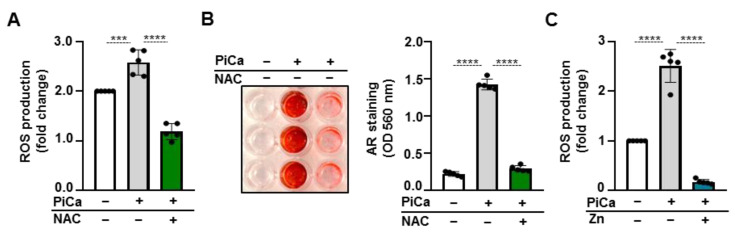
**Zinc suppresses ROS production in BMSCs.** Confluent BMSCs grown in 96-well plates were treated with Ctrl or osteogenic medium in the presence or absence of NAC (0.5 mmol/L) and zinc (50 µmol/L) for 3 days. (**A**,**C**) ROS levels were detected for 30 min, n = 5. (**B**) Representative AR staining (day 5) and quantification. Data are expressed as the mean ± SD; two other experiments yielded similar results. *** *p *< 0.005, **** *p *< 0.001.

## Data Availability

The original contributions presented in the study are included in the article; further inquiries can be directed to the corresponding author.

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
