# Peer review of "Zinc Ameliorates High Pi and Ca-Mediated Osteogenic Differentiation of Mesenchymal Stem Cells"

_nutrients, 2024, doi:10.3390/nu16234012_

Round 1

Reviewer 1 Report

Comments and Suggestions for Authors

It is an interesting study with potential implication of zinc supplementation in clinics. The reviewer has some suggestions to further improve the merit and clarification.

1.          In fact, zinc effect on calcification has been reported in the literature. For instance, the reference # 20, that showed ZnSO4 ameliorated vascular calcification in mice with chronic renal failure and mice with cholecalciferol overload. The authors may provide a little more at the Introduction, compared to this Rwf. 20, what is new and important should be better to introduce.

2.          Although the authors have provided definitions (or full spellings) of abbreviations at the end of the manuscript, the abstract should be avoiding the abbreviations without definition since it appears sometime without full manuscript provided.

3.          The authors should provide an illustrative summary of the findings to help the readers to understand the results and underlying mechanisms.

4.          Is it possible to add some in vivo model to show the translational potential? If not, the authors may add the discussion for the clinical relevance with a section of the limitation for the lack of in vivo proof.

Reviewer 2 Report

Comments and Suggestions for Authors

Review: Zinc ameliorates high Pi and Ca-mediated osteogenic differentiation of mesenchymal stem cells

1.     On line 203, “like vise” should be likewise.

2.     The legend of Figure 1 indicates that the figures show three independent experiments. But, the figures show 4-5 data points. This should be clarified.

3.     Figure 4 legend is missing the used statistical analysis.

4.     In Figures 6B and 6C, all of the addition indicators are negative for NAC; the last NAC ones should be +.

5.     Suggestions for discussion:

-        Why PiCa does not induce RUNX2 in protein level? (Figure 2B)

-        It is nicely summarized that various studies have contradictory results about Zn and osteogenesis. However, more discussion is needed especially about Park et al., 2016 (reference 19) as this paper also uses hBMCSs that showed opposite effects. The same goes for Kwun et al., 2009 (reference 16). Do the authors think this contradiction is because of dose and time? Or could the induction of osteogenic differentiation of BMSCs by high PiCa affect this? Or any other mechanism?

Reviewer 3 Report

Comments and Suggestions for Authors

The manuscript titled “Zinc ameliorates high Pi and Ca-mediated osteogenic differentiation of mesenchymal stem cells” reports zinc as an essential element that affects the differentiation process of BMSCs into osteoblasts, the cells responsible for bone tissue formation. However, the authors should address the following questions:

  1. Why were specific concentrations of phosphate (Pi) and calcium (Ca) chosen to induce osteogenic differentiation? Were different concentrations tested?
  2. Were other osteogenic differentiation markers, besides RUNX2, SOX9, OCN, and ALP, evaluated? Which ones might be relevant, and why?
  3. What are the reasons for selecting Pi and Ca instead of other conventional osteogenic inducers, such as osteogenic medium enriched with dexamethasone and vitamin D?
  4. Were there any observed differences in the response to zinc levels among different BMSC lines? Were replication experiments conducted with cells from different donors?
  5. Are there other trace elements besides zinc that have a similar effect on BMSC differentiation? How could their effects be compared to those of zinc?
  6. How do lower zinc concentrations (within physiological ranges) affect osteogenic differentiation? Was any difference in response measured?
  7. Is the inhibitory effect of zinc on osteogenic differentiation reversible? What happens if zinc is removed after a certain period?
  8. How does zinc influence the osteogenic differentiation of BMSCs in the presence of other minerals, such as magnesium or copper, which also affect bone health?
  9. Does zinc affect other phosphate transporters besides PiT-1 and PiT-2, which may be involved in the mineralization process?
  10. Could other antioxidant agents (besides NAC) counteract the effect of zinc on osteogenic differentiation? What happens when using antioxidants with different potencies?
  11. What is the exact molecular mechanism by which zinc reduces ROS production during osteogenic differentiation?
  12. Has it been assessed whether zinc affects other signalling pathways involved in osteogenic differentiation, such as the Wnt pathway, BMP pathway, or insulin-like growth factor (IGF)?
  13. Given that zinc inhibits the osteoblastic trans differentiation of VSMCs, were specific factors evaluated that could explain the differences in response between BMSCs and VSMCs?
  14. How do the expression levels of the genes RUNX2, SOX9, and ALP in VSMCs compared to those in BMSCs during osteogenic differentiation?
  15. Does zinc produce similar effects in mature bone cells, such as osteoblasts and osteocytes, in terms of inhibiting mineralization and reducing ROS?
  16. Could variations in patients' zinc nutritional status affect bone regeneration outcomes? Should zinc supplementation or monitoring be considered in certain bone treatments?

Comments on the Quality of English Language

The authors should be mindful of the writing style, as some sections are written in British English while others are in American English. Please standardize the language throughout the manuscript.

Round 2

Reviewer 3 Report

Comments and Suggestions for Authors

The authors of the manuscript "Zinc ameliorates high Pi and Ca-mediated osteogenic differentiation of mesenchymal stem cells" have made the pertinent changes suggested by the reviewers. The manuscript can be published in its current form on the journal's platform.